# FMMformer: Efficient and Flexible Transformer via Decomposed Near-field and Far-field Attention

**Tan M. Nguyen**
Department of Mathematics
University of California, Los Angeles
Los Angeles, CA, USA

**Vai Suliafu** [*]
School of Computing
Scientific Computing and Imaging (SCI) Institute
University of Utah, Salt Lake City, UT, USA

**Stanley J. Osher**
Department of Mathematics
University of California, Los Angeles, Los Angeles, CA, USA

**Long Chen**
Department of Mathematics
University of California, Irvine
Irvine, CA, USA

**Bao Wang** [†]
Department of Mathematics
Scientific Computing and Imaging (SCI) Institute
University of Utah, Salt Lake City, UT, USA

## Abstract

We propose FMMformers, a class of efficient and flexible transformers inspired by the celebrated fast multipole method (FMM) for accelerating interacting particle simulation. FMM decomposes particle-particle interaction into near-field and far-field components and then performs direct and coarse-grained computation, respectively. Similarly, FMMformers decompose the attention into near-field and far-field attention, modeling the near-field attention by a banded matrix and the far-field attention by a low-rank matrix. Computing the attention matrix for FMM-formers requires linear complexity in computational time and memory footprint with respect to the sequence length. In contrast, standard transformers suffer from quadratic complexity. We analyze and validate the advantage of FMMformers over the standard transformer on the Long Range Arena and language modeling benchmarks. FMMformers can even outperform the standard transformer in terms of accuracy by a significant margin. For instance, FMMformers achieve an average classification accuracy of $60.74\%$ over the five Long Range Arena tasks, which is significantly better than the standard transformer's average accuracy of $58.70\%$.

## 1 Introduction

Transformers [58] have achieved state-of-the-art performance in sequence processing tasks, including machine translation and language modeling [58, 2, 15, 4, 61, 16, 9]. Also, transformers can effectively transfer knowledge from a pre-trained model to tasks with limited supervision [43, 44, 16, 64, 34]. Transformers rely on the attention mechanism and particularly self-attention as a fundamental building block for their modeling [5, 58, 27].

### 1.1 Self-attention

The self-attention mechanism is used to learn long-range dependencies while enabling parallel processing of the input sequence. For a given input sequence $\boldsymbol{X} := [\boldsymbol{x}_1, \boldsymbol{x}_2, \cdots, \boldsymbol{x}_N]^\top \in \mathbb{R}^{N \times D_x}$ of

---

[*]Equal contribution and Co-first author
[†]Please correspond to: wangbaonj@gmail.com or chenlong@math.uci.edu

35th Conference on Neural Information Processing Systems (NeurIPS 2021).

$N$ feature vectors that have been encoded in a $D_x$-dimensional vector space, self-attention transforms $\boldsymbol{X}$ into an output sequence $\hat{V}$ in the following two steps:

Step 1. Project the input sequence $\boldsymbol{X}$ into three matrices via the following linear transformations

$$\boldsymbol{Q} = \boldsymbol{X}\boldsymbol{W}_Q^\top; \boldsymbol{K} = \boldsymbol{X}\boldsymbol{W}_K^\top; \boldsymbol{V} = \boldsymbol{X}\boldsymbol{W}_V^\top,$$

where $\boldsymbol{W}_Q, \boldsymbol{W}_K \in \mathbb{R}^{D \times D_x}$, and $\boldsymbol{W}_V \in \mathbb{R}^{D_v \times D_x}$ are the weight matrices. We denote $\boldsymbol{Q} := [\boldsymbol{q}_1, \cdots, \boldsymbol{q}_N]^\top, \boldsymbol{K} := [\boldsymbol{k}_1, \cdots, \boldsymbol{k}_N]^\top$, and $\boldsymbol{V} := [\boldsymbol{v}_1, \cdots, \boldsymbol{v}_N]^\top$, where the vectors $\boldsymbol{q}_i, \boldsymbol{k}_i, \boldsymbol{v}_i$ for $i = 1, \cdots, N$ are the query, key, and value vectors, respectively.

Step 2. For each query vector $\boldsymbol{q}_i$ for $i = 1, \cdots, N$, we compute the output vector $\hat{\boldsymbol{v}}_i$ as follows

$$\hat{\boldsymbol{v}}_i = \sum_{j=1}^{N} \mathrm{softmax}\Big(\frac{\boldsymbol{q}_i^\top \boldsymbol{k}_j}{\sqrt{D}}\Big)\boldsymbol{v}_j, \iff \hat{V} = \Big(\frac{\boldsymbol{Q}\boldsymbol{K}^\top}{\sqrt{D}}\Big)\mathbf{V} := \boldsymbol{A}\boldsymbol{V}, \tag{1}$$

where the softmax function is applied to each row of the matrix $(\boldsymbol{Q}\boldsymbol{K}^\top)/\sqrt{D}$.

For long sequences, the computational time and memory footprint of transformers are dominated by (1). It is evident that the memory cost is $\mathcal{O}(N^2)$ to store the attention matrix $\boldsymbol{A}$. Also, the computational complexities of computing the matrix-matrix products $\boldsymbol{Q}\boldsymbol{K}^\top$ and $\boldsymbol{A}\boldsymbol{V}$ are both $\mathcal{O}(N^2)$. These limitations impede the application of transformers to many important settings that involve very long sequences [33, 25, 39]. When applying self-attention for long sequence modeling, we have to limit the context window to a reasonable size to make it computationally feasible, limiting the effectiveness of learning long-range dependencies. Efficient transformer models have been proposed, including leveraging sparse and low-rank attention. Many of the existing efficient transformers gain computational and memory efficiency at the cost of significant accuracy degradation.

## 1.2 Contribution

Leveraging the idea of the fast multipole method (FMM) [19], we propose a class of efficient, flexible, and expressive transformers, namely *FMMformers*. At the core of FMMformers is to replace the self-attention $\hat{V} = \boldsymbol{A}\boldsymbol{V}$ in (1) with the following matrix-matrix product

$$\hat{V} := (\boldsymbol{D} + \boldsymbol{L})\boldsymbol{V}, \tag{2}$$

where $\boldsymbol{D}$ is a banded matrix with bandwidth $k \ll N$ and $\boldsymbol{L}$ is a low-rank matrix of rank $r \ll N$. In practice, we normalize matrix $\boldsymbol{D} + \boldsymbol{L}$ such that the sum of each row is 1; for the sake of presentation, we ignore this normalization step below. Both $\boldsymbol{D}\boldsymbol{V}$ and $\boldsymbol{L}\boldsymbol{V}$ can be computed with linear computational and memory complexity; they model the near-field and far-field attention, respectively. FMMformers are flexible in designing the sparse banded matrix and the low-rank matrix for modeling near-field and far-field attention. In particular, we can control the bandwidth of the banded matrix $\boldsymbol{D}$ and the rank of the low-rank matrix $\boldsymbol{L}$ for expressivity and efficiency tradeoff. In addition to the efficiency and flexibility, FMMformers gain significant accuracy improvement over linear transformers and can even outperform the standard transformer in terms of accuracy. We illustrate the idea of FMMformers in Figure 1: Instead of modeling the full attention by a dense unstructured matrix, we employ a sparse banded matrix to model the near-field attention and several rank one matrices to model the far-field attention.

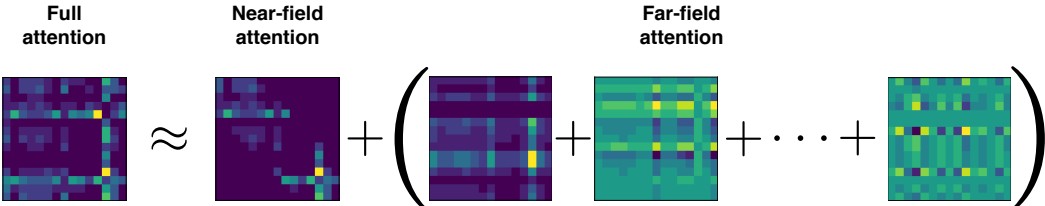

**Full attention**   **Near-field attention**   **Far-field attention**

Figure 1: Left-hand side: we visualize a randomly selected full attention map (the matrix $\boldsymbol{A}$ in (1)) from the standard transformer trained for the CIFAR10 image classification task in the Long Range Arena (LRA) benchmark. Right-hand side: we illustrate how this attention map can be decomposed into near-field and far-field attention, which are modeled by a sparse banded matrix and the sum of several rank one matrices in our FMMformer, respectively.

## 1.3 Organization

We structure this paper as follows: In Sec. 2, we briefly review the celebrated FMM and establish the connection between FMM and self-attention. In Sec. 3, we present a practical implementation of FMMformer that leverages existing techniques for low-rank approximation of the self-attention mechanism. We validate and empirically analyze the efficiency and accuracy of FMMformers in Sec. 4. We discuss related works in Sec. 5. The paper ends up with concluding remarks. Technical proofs and more experimental details are provided in the Appendix.

## 2 Fast Multipole Method and Self-attention Mechanism

In this section, we review FMM and present an algebraic interpretation of FMM, see Sec. 2.1. Then, in Sec. 2.2, we explore the structure of the attention matrix, showing that FMM can be used to accelerate the self-attention mechanism.

### 2.1 Fast multipole method vs. sparse and low-rank matrix approximation

FMM is a numerical method that was originally developed to speed up the calculation of long-range forces in the $n$-body problem [19] and has been regarded as one of the top 10 algorithms in scientific computing in the 20th century [14]. The key idea is that *the far-field interaction can be well-approximated by separable low-rank matrices*, while the near-field interaction can be calculated directly. We use the following simple example to illustrate mathematical reasoning. Without ambiguity, we reuse notations in the previous section and assume:

(A1) $\boldsymbol{A}(i,j) = g(|\boldsymbol{q}_i - \boldsymbol{k}_j|)$ depends on the distance of two vectors $\boldsymbol{q}_i$ and $\boldsymbol{k}_j$, where $\boldsymbol{A}(i,j)$ is the $(i,j)$-th entry of the matrix $\boldsymbol{A} \in \mathbb{R}^{N \times N}$.

(A2) The function $g(s)$ is smooth for $s \neq 0$.

(A3) The function $g$ satisfies $g(st) = g(s)g(t)$.

One noticeable example in the physical application is the gravitational potential $g(|\boldsymbol{q}_i - \boldsymbol{k}_j|) = 1/|\boldsymbol{q}_i - \boldsymbol{k}_j|$, for which the key vectors $\{\boldsymbol{k}_j\}$ are the location of source particles and the query vectors $\{\boldsymbol{q}_i\}$ are the location of the target points. Assumption (A3) is not essential, which is presented here for the convenience of proof and can be replaced by other separable forms, e.g., $g(st) = g(s) + g(t)$. The near-field and far-field are defined through the distance $|\boldsymbol{q}_i - \boldsymbol{k}_j|$.

We now explain the low-rank approximation based on the well-separated condition. For illustrative purpose, we assume the index set $\{1, 2, \ldots, N\}$ is partitioned into two groups $\{T_1, T_2\}$.

**Definition 1.** *Group $T_1$ is called well-separated from $T_2$ if there exists a vector $\boldsymbol{k}^*$ and a number $\delta \in (0, 1)$ such that*
$$|\boldsymbol{k}_j - \boldsymbol{k}^*| \leq \delta|\boldsymbol{q}_i - \boldsymbol{k}^*| \quad \forall i \in T_1, j \in T_2.$$

The vector $\boldsymbol{k}^*$ is a representative vector of $\{\boldsymbol{k}_j, j \in T_2\}$, e.g., the center of vectors in $T_2$. For any $\boldsymbol{q}_i, i \in T_1$, it is far away from $\{\boldsymbol{k}_j, j \in T_2\}$ and the far-field interaction $A(i,j), i \in T_1, j \in T_2$ can be approximated by $g(|\boldsymbol{q}_i - \boldsymbol{k}^*|)$, i.e. each row of $A(T_1, T_2)$, the submatrix of $\boldsymbol{A}$ with the row index set $T_1$ and the column index set $T_2$, is constant. For example, when calculating the gravitation of a galaxy from the Earth, we can simply treat the galaxy as one single point, although the galaxy may contain hundreds of millions of stars. By including $p$ terms of the Taylor series, the approximation can be more accurate by using rank $p$ instead of rank 1 matrix approximation.

**Lemma 1.** *Let $\{T_1, T_2\}$ be two well-separated index sets. Assume (A1)-(A3) hold. For any $\varepsilon > 0$, the sub-matrix $A(T_1, T_2)$ can be uniformly approximated by a rank $p$ matrix to a tolerance $\varepsilon > 0$ in the sense that: there exists rank $p$ matrices $\boldsymbol{U} \in \mathbb{R}^{|T_1| \times p}, \boldsymbol{V} \in \mathbb{R}^{|T_2| \times p}$, with $p \geq C|\log_\delta \epsilon|$ for some positive constant C, such that*
$$|\boldsymbol{A}(i,j) - (\boldsymbol{U}\boldsymbol{V}^\top)(i,j)| \leq \epsilon, \quad \forall i \in T_1, j \in T_2.$$

The applicability of the analytic kernel function $g$ was limited to partial differential equations or integral equations where Green's function satisfying (A1)-(A3). In the application of machine learning, it is hard to verify (A1)-(A3). Instead, we use the definition of diagonal-plus-semi-separable matrices from the book [6, Definition 1.10]. We use MATLAB/Numpy notation $\mathrm{tril}(\boldsymbol{K}, p)$ to denote the lower triangular matrix with zeros above the $p$th-subdiagonal of $\boldsymbol{K}$ and similar notation $\mathrm{triu}(\boldsymbol{K}, p)$ for the upper triangular part.

**Definition 2.** *[6, Definition 1.10] A matrix $\boldsymbol{A} \in \mathbb{R}^{N \times N}$ is called $(p,q)$-semi-separable if there exist matrices $\boldsymbol{U}, \boldsymbol{V} \in \mathbb{R}^{N \times p}$ and $\boldsymbol{W}, \boldsymbol{Z} \in \mathbb{R}^{N \times q}$ such that*

$$\boldsymbol{A} = \mathrm{triu}(\boldsymbol{U}\boldsymbol{V}^\top, 0) + \mathrm{tril}(\boldsymbol{W}\boldsymbol{Z}^\top, 1).$$

*It is called diagonal-plus-semi-separable if*

$$\boldsymbol{A} = \boldsymbol{D} + \mathrm{triu}(\boldsymbol{U}\boldsymbol{V}^\top, 1) + \mathrm{tril}(\boldsymbol{W}\boldsymbol{Z}^\top, 1).$$

*with some diagonal matrix $\boldsymbol{D}$.*

Definition 2 can be naturally extended to include a banded matrix $\boldsymbol{D}$ and sum of several low-rank matrices. Moreover, One can verify the semi-separable property of matrix $\boldsymbol{K}$ by checking the decay of singular values of the matrix. As often used in low-rank approximation methods, the numerical rank or $\varepsilon$-rank of a matrix $\boldsymbol{K}$, for a tolerance $\varepsilon$, is the number of singular values of $\boldsymbol{K}$ that are greater than $\varepsilon\|\boldsymbol{K}\|_2$.

The low rank approximation relies on the well separateness of two subsets. Based on a hierarchical partition of the index set, a $\mathcal{H}$-matrix [21] can be constructed; see Figure 2 for an illustration. Further compression leads to $\mathcal{H}^2$-matrix [20, 22] and the hierarchically semi-separable (HHS) matrix [11, 62]. Other variants include hierarchically block-separable (HBS) [36], and hierarchically off-diagonal low-rank (HODLR) [3] matrices, etc.

In our application, we write the decomposition as

$$\boldsymbol{A} = \boldsymbol{D} + \sum_{l=1}^{r} \phi_l(\boldsymbol{Q})\phi_l^\top(\boldsymbol{K}). \qquad (3)$$

In the query and key spaces, the vectors $\boldsymbol{q}_i$ and $\boldsymbol{k}_j$ may not be well-separated. Then nonlinear feature maps $\phi_l(\cdot), l = 1, \cdots, r$ to higher dimensions, which are trainable, can be used to make the mapped datasets more separable. In (3), each kernel function $\phi_l : \mathbb{R}^N \to \mathbb{R}^N$ is a vector function of length $N$. We can mimic the hierarchical decomposition used in $\mathcal{H}$-matrix to use low rank kernel approximation $\phi_l(\boldsymbol{Q}(1 : N/2, :)\phi_l^\top(\boldsymbol{K}(N/2+1 : N, :))$ with halved length. Such approximation can be recursively applied to get a multilevel decomposition in the low rank approximation component. Our numerical results show that a simple one level near-field and far-field decomposition is good enough.

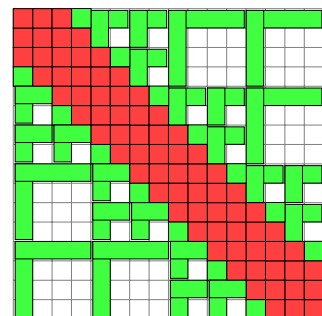

Figure 2: A $\mathcal{H}$-matrix based on a hierarchical decomposition of the index set. The red part is a banded matrix and the green part can be written as sum of low rank matrices.

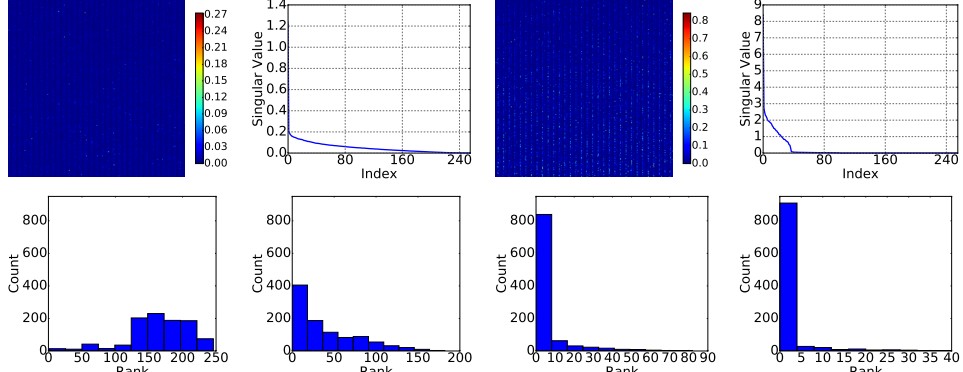

Figure 3: First row: plot of two randomly selected attention matrices (left) and their singular values (right) from the transformer trained for WikiText-103 language modeling; see Sec. 4.3 for details. Second row: distributions of the rank of randomly selected 1000 attention matrices, from the same transformer, after removing a banded matrix $\boldsymbol{D}$ of bandwidth 0 (not remove anything from the matrix $\boldsymbol{A}$), 5, 10, and 20 (from left to right). Matrix $\boldsymbol{A} - \boldsymbol{D}$ is of low rank, and the rank becomes smaller in general when the bandwidth of $\boldsymbol{D}$ increases.

## 2.2 Sparse and low-rank patterns in attention maps

In this section, we explore the sparse and low-rank structure of the attention matrix $\boldsymbol{A}$. In particular, we consider the attention matrix $\boldsymbol{A} \in \mathbb{R}^{256 \times 256}$ obtained from the standard transformer trained for

WikiText-103 language modeling; see Sec. 4.3 for the experimental details. We randomly select 1000 different attention matrices, and we exclude a banded matrix $D$ with bandwidth 5, 10, and 20 from each of such matrices. Then, we perform singular value decomposition (SVD) to compute the rank of each matrix $A - D$, and we threshold the small singular values with a magnitude of $10^{-6}$. Figure 3 (top row) plots two randomly selected self-attention matrices and the distribution of the rank of the matrix $A - D$. It is clear that matrix $A$ has only a few large singular values and all other singular values are very small. Moreover, matrix $A - D$ is of low rank, and the rank becomes smaller in general when the bandwidth of $D$ increases, which is consistent with the assumptions in Sec. 2.1, motivating FMMformers.

# 3 FMMformer: Practical Near-field and Far-field Attention

In this section, we present practical algorithms for implementing the proposed FMMformer defined by (2). In particular, we present fast algorithms for computing the near-field attention $DV$ and the far-field attention $LV$.

## 3.1 Banded matrix modeling of near-field attention

We model the near-field attention with the following banded matrix

$$D = \mathrm{softmax}\left(\mathrm{band}_k\left(\frac{QK^\top}{\sqrt{D}}\right)\right), \tag{4}$$

where the operator $\mathrm{band}_k(*)$ represents taking only the banded part of the matrix $*$ with a bandwidth $k$ ($k \ll N$). In practice, there is no need to calculate the matrix product $QK^\top$. Instead, we only need to calculate the products of the vectors that correspond to the nonzero entries of the banded matrix $\mathrm{band}_k(QK^\top/\sqrt{D})$. Note that for long sequences, both the time and memory complexity of computing (4) are $\mathcal{O}(kN)$.

## 3.2 Low-rank matrix modeling of far-field attention

We consider practical and efficient low-rank matrix modeling of the far-field attention $LV$ in (1). In principle, any existing off-the-shelf low-rank attention can be integrated into FMMformer to model the far-field attention. In particular, we model the far-field attention by leveraging the kernel trick used in [26, 13, 47], which is flexible in selecting different kernels to modulate the rank of the far-field attention.

### 3.2.1 Low-rank attention via kernelization

Suppose we model the far-field attention using a rank $r$ matrix $L \in \mathbb{R}^{N \times N}$, which can be written as the sum of $r$ rank one matrices, i.e.,

$$L = a_1 b_1^\top + a_2 b_2^\top + \cdots + a_r b_r^\top, \tag{5}$$

where $a_1, a_2, \cdots, a_r; b_1, b_2, \cdots, b_r \in \mathbb{R}^N$. Note that

$$LV = (a_1 b_1^\top + a_2 b_2^\top + \cdots + a_r b_r^\top)V = a_1(b_1^\top V) + a_2(b_2^\top V) + \cdots + a_r(b_r^\top V), \tag{6}$$

which indicates that we can compute $LV$ with $\mathcal{O}(rN)$ time complexity. Also, we only need to store the vectors $\mathbf{u}_1, \cdots, \mathbf{u}_r; \mathbf{v}_1, \cdots, \mathbf{v}_r \in \mathbb{R}^N$, resulting in linear complexity in memory footprint.

We borrow the idea of kernelization from the linear transformer [26] for practical implementation of (6). In particular, the authors in [26] generalize the softmax function in (1) to a general kernel function $k(q_i, k_j)$, i.e.,

$$\underbrace{\hat{v}_i = \frac{\sum_{j=1}^N \exp(q_i, k_j)v_j}{\sum_{j=1}^N \exp(q_i, k_j)}}_{\text{self-attention}} {}^3 \implies \underbrace{\hat{v}_i = \frac{\sum_{j=1}^N k(q_i, k_j)v_j}{\sum_{j=1}^N k(q_i, k_j)}}_{\text{generalized self-attention}}. \tag{7}$$

If $k(q_i, k_j) = \phi(q_i)^\top \phi(k_j)$ for a certain feature map $\phi(\cdot)$, then we have

$$\hat{v}_i = \frac{\sum_{j=1}^N k(q_i, k_j)v_j}{\sum_{j=1}^N k(q_i, k_j)} = \frac{\sum_{j=1}^N \phi(q_i)^\top \phi(k_j)v_j}{\sum_{j=1}^N \phi(q_i)^\top \phi(k_j)} = \frac{\phi(q_i)^\top \sum_{j=1}^N \phi(k_j)v_j^\top}{\phi(q_i)^\top \sum_{j=1}^N \phi(k_j)}, \tag{8}$$

---

$^3$Here, $\exp(q_i, k_j) := \exp(q_i^\top k_j/\sqrt{D})$.

Note that (8) can be regarded as a rank one approximation of self-attention. We can rewrite (8) into the following compact form

$$\hat{V} = \frac{\phi(Q)(\phi(K)^\top V)}{\phi(Q)\phi(K)^\top}. \tag{9}$$

To generalize (8) to the rank $r$ approximation, we select a set of linearly independent feature maps $\{\phi_l(\cdot)\}_{l=1}^r$. Together with the sparse banded matrix modeling of the near-field attention, we propose the following efficient attention model for the FMMformer

$$\hat{V} = DV + \sum_{l=1}^r \frac{\phi_l(Q)(\phi_l(K)^\top V)}{\phi_l(Q)\phi_l(K)^\top}. \tag{10}$$

It is evident that both computational time and memory complexity are linear in computing (10). Our design is flexible to selecting feature maps and the sparse banded matrix, which the users can customize. Moreover, causal masking can be implemented easily by truncating the sum from 1 to $i$ in (8) together with masking out the corresponding part of the banded matrix $\mathbf{D}$.

**Proposition 1.** *Let $\phi_l(\boldsymbol{x}) \in \mathbb{R}^N$ ($l = 1, 2, \cdots, r$ and $r \ll N$) for $\boldsymbol{x} \in \mathbb{R}^n$. If $\{\phi_l(\boldsymbol{x})\}_{l=1}^r$ are linearly independent at $\boldsymbol{x}$, then the following matrix $\boldsymbol{L}(\boldsymbol{x}) \in \mathbb{R}^{N \times N}$ has rank $r$,*

$$\boldsymbol{L}(\boldsymbol{x}) := \phi_1(\boldsymbol{x})\phi_1(\boldsymbol{x})^\top + \phi_2(\boldsymbol{x})\phi_2(\boldsymbol{x})^\top + \cdots + \phi_r(\boldsymbol{x})\phi_r(\boldsymbol{x})^\top. \tag{11}$$

**Feature map selection.** The feature map selection is crucial for the success of far-field attention modeling. In this work, we adopt the existing successful feature map $\phi_1(\boldsymbol{x}) := \mathrm{elu}(\boldsymbol{x}) + 1$ used in the linear transformer [26] together with $\phi_2(\boldsymbol{x}) := \mathrm{elu}(-\boldsymbol{x}) + 1$, which is a straightforward modification of $\phi_1(\boldsymbol{x})$. Moreover, we consider the third feature map $\phi_3(\boldsymbol{x}) := \tanh(\boldsymbol{x})$. It is easy to check that $\phi_1(\boldsymbol{x}), \phi_2(\boldsymbol{x})$, and $\phi_3(\boldsymbol{x})$ are linearly independent for almost all $\boldsymbol{x}$. We leave how to design a set of feature maps to optimize the far-field attention modeling as future work.

### 3.3 Blending of near-field and far-field attention

Based on our experiments, adding a learnable weight in front of each attention component benefits training and generalization. As such, we propose the following scheme to blend the near-field attention and far-field attention

$$\hat{V} := (w_1 D + w_2 L)V, \tag{12}$$

where $w_1$ and $w_2$ are two learnable weights, and we enforce their positivity via a sigmoid map.

## 4 Experimental Results

In this section, we numerically verify the efficiency of FMMformers and empirically analyze the effects of near-field and far-field attention on various benchmarks, including synthetic sequence copy (Sec. 4.1), Long Range Arena (LRA) (Sec. 4.2), and language modeling (Sec. 4.3). We aim to show that: (i) FMMformers are efficient in both computational time and memory footprint. (ii) Multiple kernels benefit learning of the far-field attention. (iii) Blending near-field attention with far-field attention can boost the performance of linear transformers. Throughout this section, we compare FMMformers with linear transformers (**linear**, $r = 1$ in (11)), standard softmax transformers (**softmax**), and softmax transformers that use a banded attention matrix of bandwidth $k$ (**band**$_k$). All experiments are conducted on a server with 4 NVIDIA 3090TI GPUs.

### 4.1 Synthetic sequence copy task

We first consider a synthetic copy task with various sequence lengths, including 128, 256, and 512. In this task, the model has to duplicate a sequence of symbols. Each training and test sample is a sequence of maximum length 128/256/512 with ten different symbols separated by a dedicated separator symbol. We train all transformers for this task using the same setting as in [26].

**Boosting performance of linear transformers with near-field attention.** We first compare FMM-formers, obtained by blending the linear transformer with a banded attention matrix of bandwidths 10, 20, and 30, respectively. Figure 4 shows that for shorter sequences of length 128, all transformers reach similar loss; the standard softmax transformer converges much faster than the linear transformer while *blending the linear transformers with near-field attention can improve training*. Moreover, the benefits of near-field attention become more significant as the sequence length increases.

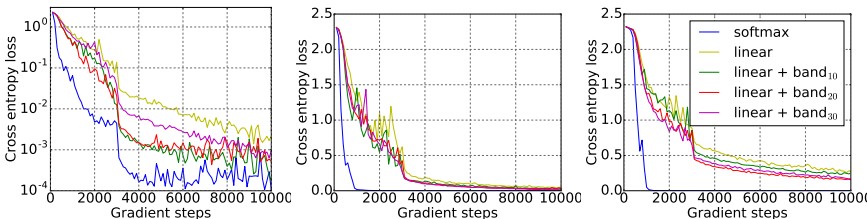

Figure 4: Convergence comparison of softmax, linear, and the blend of linear transformer with a banded matrix on a sequence duplication task with different sequence lengths (left: 128, middle: 256, right: 512). Adding near-field attention into linear attention consistently improves the training for different sequence lengths.

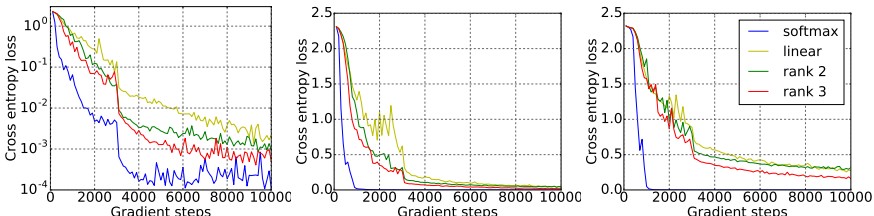

Figure 5: Convergence comparison of softmax, linear, and different low-rank attention on a sequence duplication task with different sequence lengths (left: 128, middle: 256, right: 512). Attention with a higher rank improves training for different sequence lengths.

**Enhancing far-field attention with multi-kernels.** After observing that the linear transformer performs poorly as the sequence length increases, we consider augmenting the linear transformer with multiple feature maps; in particular, we consider the three feature maps mentioned above, i.e., $\phi_1(\boldsymbol{x}) = \text{elu}(\boldsymbol{x}) + 1, \phi_2(\boldsymbol{x}) = \text{elu}(-\boldsymbol{x}) + 1$, and $\phi_3(\boldsymbol{x}) = \text{tanh}(\boldsymbol{x})$. Figure 5 compares different transformers on different sequence lengths, where **rank 2** consists of the feature maps $\phi_1(\boldsymbol{x})$ and $\phi_2(\boldsymbol{x})$, and **rank 3** consists of all three feature maps. These results show that *multiple kernels can improve the learning of far-field attention.*

**Computational and memory complexity.** In this part, we compare different transformers in computational time and memory cost. Following [26], we compute the attention and gradient for input sequences with different lengths $N \in \{2^9, 2^{10}, \cdots, 2^{16}\}$ and measure the peak allocated GPU memory and the required time for each transformer model. We conduct this experiment on an NVIDIA 3090TI with 24GB memory, and we report the time and memory cost per sample in the same way as in [26]. Figure 6 contrasts the time (left) and memory (right) costs of different models.

## 4.2 Long Range Arena (LRA) Benchmark

In this experiment, we evaluate our model on tasks that involve longer sequence lengths in the Long Range Arena benchmark [54]. We show that the FMMformer outperforms the baseline linear transformer and standard softmax transformer [58], justifying the advantage of the FMMformer in capturing long-range dependencies. We provide model and training details in the Appendix.

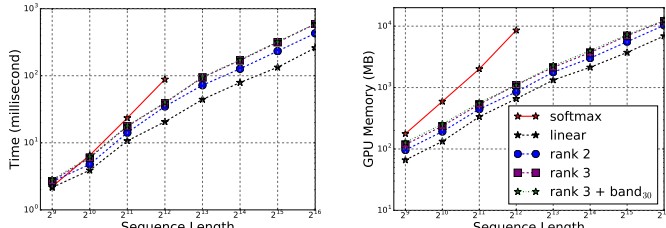

Figure 6: Comparison of the computational time and the peak memory cost of a forward/backward pass for standard softmax transformer, linear transformer, rank 2 linear transformer, rank 3 linear transformer, and the blend of rank 3 linear transformer with a banded attention matrix of bandwidth 30. All transformers are of linear complexity in time and memory except the softmax transformer.

| Model | ListOps (2K) | Text (4K) | Retrieval (4K) | Image (1K) | Pathfinder (1K) | Avg |
|---|---|---|---|---|---|---|
| Softmax [58] | **37.10 (37.10)** | 64.17 (65.02) | 80.71 (79.35) | 39.06 (38.20) | **72.48 (74.16)** | 58.70 (58.77) |
| Linear [26] | 18.30 | 64.22 | 81.37 | 38.29 | 71.17 | 54.67 |
| Band$_5$ | 32.16 | 66.31 | 79.41 | 43.33 | 67.44 | 57.73 |
| FMMformer (1-kernel + Band$_5$) | 33.22 | 66.52 | 81.50 | 45.01 | 71.29 | 59.51 |
| FMMformer (2-kernel + Band$_5$) | 36.74 | **67.84** | **81.88** | **45.10** | 72.12 | **60.74** |

Table 1: Results on the LRA benchmark. We report the test classification accuracy for each task and average accuracy across all tasks. The FMMformer outperforms the linear transformer and attains similar or better results than the standard transformer. Across tasks, the FMMformer achieves the best average accuracy. Also, the FMMformer with 2 kernels enhances the performance of the FMMformer with 1 kernel. The numbers in the parenthesis are from the paper [63]. Note that we use near-field attentions of bandwidth 5 for all FMMformers reported here, and Band$_5$ are softmax transformers with a banded attention matrix of bandwidth 5.

**Datasets and metrics.** We consider all five tasks in the LRA benchmark, including Listops [38], byte-level IMDb reviews text classification [35], byte-level document retrieval [42], CIFAR-10 image classification on sequences of pixels [30], and Pathfinder [32]. These tasks involve long sequences of length $2K$, $4K$, $4K$, $1K$, and $1K$, respectively. We follow the setup/evaluation protocol in [54] and report the test accuracy for individual task and the average result across all tasks.

**Results.** We summarize our results in Table 1. Like in the copy task, we observe that adding near-field attention modeled by banded attention matrices improves the performance of linear transformers. More interestingly, using bandwidth 5 already yields good results across all LRA tasks while significantly reducing the computational and memory cost of calculating the attention matrix. For example, in the byte-level document retrieval [42] task, a banded matrix with bandwidth 5 only accounts for 0.125% of the corresponding full attention matrix. The FMMformer with 1 kernel (blending a banded matrix of bandwidth 5 with the linear transformer using feature map $\phi_1(\boldsymbol{x})$) outperforms the linear transformer and yields similar or better results than the standard softmax transformer in all tasks. Furthermore, the FMMformer with 2 kernels (blending a banded attention matrix of bandwidth 5 with the linear transformer using feature maps $\phi_1(\boldsymbol{x})$ and $\phi_2(\boldsymbol{x})$) further improves the FMMformer with 1 kernel, justifying the need of better low-rank approximation for the far-field attention. Across tasks, the FMMformer obtains the best average accuracy. Also, it is worth noting that tasks in the LRA benchmark cover different data modalities include text and images. Good performance of the FMMformer on these tasks demonstrates that the advantages of our model over the linear and standard transformers are consistent across data modalities.

### 4.3 Language Modeling on WikiText-103

Experiments on the copy task in Sec. 4.1 illustrate the effect of combining near-field and far-field attention. Results on the LRA benchmark in Sec. 4.2 show the ability of our FMMformer to capture very long-range dependency and extend to different data modalities. Now our goal is to confirm the advantage of the FMMformer on a large-scale application. We consider the word-level language modeling task on WikiText-103 [37].

**Datasets and metrics.** WikiText-103 consists of articles from Wikipedia and is a dataset with long contextual dependencies. The training set is made up of about $28K$ articles containing $103M$ running words; this corresponds to text blocks of about 3600 words. The validation and test sets are composed of $218K$ and $246K$ running words, respectively. Each of them contains 60 articles and about $268K$ words. Our experiment follows the standard setting [37, 47] and split the training data into $L$-word independent long segments. For evaluation, we use a batch size of 1, and go through the text sequence with a sliding window of size $L$. We consider only the last position for computing perplexity (PPL) except in the first segment, where all positions are evaluated as in [2, 47].

**Results.** Table 2 shows the validation and test perplexity of our models versus the linear and standard softmax transformer on WikiText-103. Here, we also compare with the linear attention with the fast weight trick proposed in [47]. Consistent with previous experiments, the FMMformer outperforms the linear transformer with or without fast weight. The standard softmax transformer obtains the best results in this task, but the gap between the FMMformer and the standard transformer is very small when a larger bandwidth is used for near-field attention in the FMMformer. This is justified by the improvement in terms of PPL of the FMMformer with a near-field attention of bandwidth 20 compared to the FMMformer with a near-field attention of bandwidth 5. Also, FMMformer with 2 kernels ($\phi_1(\boldsymbol{x})$ and $\phi_2(\boldsymbol{x})$) still improves over FMMformer with 1 kernel ($\phi_1(\boldsymbol{x})$). Consider the linear complexity of computational time and memory advantage of FMMformers, the small performance gap of FMMformers to standard softmax transformers can potentially be overcome by using the multilevel or hierarchical near-field and far-field decomposition.

| Method | Valid PPL | Test PPL |
|---|---|---|
| Softmax [58] | 33.15 | 34.29 |
| Linear [26] | 37.27 | 38.40 |
| Fast weight [47] | 35.75 | 36.63 |
| Fast weight [47] + Linear [26] | 34.78 | 35.95 |
| $Band_{20}$ | 38.18 | 39.19 |
| FMMformer (1-kernel linear + $Band_{20}$) | 35.41 | 36.43 |
| FMMformer (1-kernel fast weight + $Band_{20}$) | 34.54 | 35.47 |
| FMMformer (2-kernel linear + $Band_{20}$) | 35.10 | 36.11 |
| FMMformer (2-kernel fast weight + $Band_{20}$) | 34.16 | 34.71 |

Table 2: WikiText-103 language model perplexities of FMMformers compared to the baselines. The number of parameters (40 M) is almost the same for all models, up to the small difference introduced by additional weights on the far-field attention in FMMformers. FMMformers outperform linear transformers [26]. The performance gap compared to softmax transformers is reduced when using a larger bandwidth in near-field attention and more kernels in far-field attention. Note that $Band_5$ and $Band_{20}$ are softmax transformers with a banded attention matrix of bandwidth 5 and 20, respectively.

## 5 Related works.

**Low-rank transformers.** Low-rank approximation of the self-attention matrix $\boldsymbol{A}$ has been a popular method in reducing the quadratic computational and memory complexity of transformers to linear. Linearized attention that leverages kernelization tricks can be considered as the rank one approximation of the self-attention matrix [60, 26, 13, 49]; the choice of the feature map function is crucial for the success of linearized attention. Fast weight memories [48] have been used to improve memory capacity of linearized attention [47]. The Nyström method has also been leveraged for developing efficient attention with linear computational complexity [63]. Many other low-rank attention models exist, e.g., [8, 45, 50, 40]. FMMformers employ low-rank attention to model far-field attention; in principle, the merits of existing low-rank attention can be integrated into FMMformers.

**Sparse transformers.** Attention matrices have been enforced with different sparsity patterns to gain efficiency, including fixed sparsity patterns [41, 39, 7, 1, 65], a combination of different sparsity patterns [12, 24], and data-dependent/learnable sparsity patterns [33, 60, 52, 53, 29, 46, 59]. Informer [66] is another efficient attention scheme using a sparse query. Note that the existing sparsity pattern can be very complicated, while we adopt a sparse banded matrix to model the near-field attention.

**Other efficient transformers.** Reformer reduces the cost of self-attention to $\mathcal{O}(N \log N)$ via locality-sensitive hashing [29]. Linformer [60] and Longformer [7] obtain the linear complexity using random projection and local window attention, respectively. Galerkin transformer in [10] uses a Galerkin self-attention for the encoder. See [55] for a review of efficient transformers.

**Sparse and low-rank interpretation of FMM.** The fast multipole method is introduced by Greengard and Rokhlin [19] for the efficient computation of gravitational/electrostatic potentials and fields. Applications of FMM to machine learning can be found in [18, 31, 57]. FMM can be generalized algebraically for efficient computation of the dense matrix-vector product. Algebraic counterpart of FMM include $\mathcal{H}$-matrix [21], $\mathcal{H}^2$-matrix [20, 22], hierarchically semi-separable (HSS) [11, 62], hierarchically block-separable (HBS) [36], and hierarchically off-diagonal low-rank (HODLR) [3] matrices. The common feature is to compress the off-diagonal sub-matrices by low-rank approximations.

## 6 Concluding Remarks

In this paper, we proposed FMMformers, a class of efficient and flexible transformers with linear time and memory complexity, inspired by the fast multipole method. In FMMformers, we decompose the full attention into near-field and far-field attention; we model the near-field attention with a sparse banded matrix and model the far-field attention using a low-rank matrix leveraging ideas of the linear transformer [26]. We validate the efficiency of FMMformers on various benchmark tasks, including synthetic sequence copy, LRA benchmark, and WikiText-103 language modeling. Our numerical results show that FMMformers consistently outperform the linear transformer on all benchmarks and outperform the standard softmax transformer on the LRA tasks. In our work, we select linearly independent feature maps to enhance the learning of far-field attention. It is natural to ask how to design a set of feature maps to optimize the performance of FMMformers? Furthermore, we leave the application of FMMformers for improving the vision transformer [17, 56] as future work.

# 7 Acknowledgement

This material is based on research sponsored by NSF grants DMS-1924935, DMS-1952339 and DMS-2012465, DOE grant DE-SC0021142, and ONR grant N00014-18-1-2527 and the MURI grant N00014-20-1-2787. We thank Professor Jack Xin for helpful discussions.

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
