# OpenReview forum: "FMMformer: Efficient and Flexible Transformer via Decomposed Near-field and Far-field Attention"
_NeurIPS.cc/2021/Conference — NeurIPS 2021 Poster_

### Official Review · Reviewer_zdGe · 2021-07-15

**Rating:** 6
**Confidence:** 3

**Summary:**

The paper proposed the new efficient transformer (FMMformers), which decomposes the attention computation into direct and coarse-grained computations. The coarse-grained computation is handled by the low-rank matrix and the direct computation is handled by a banded matrix. The proposed method shows better performance on the LRA dataset and comparable performance on the language modeling with the original transformer


**Limitations And Societal Impact:**

(1) Implementation details: given that local window approach like Big-Bird or Longformer generally needs to have a customized kernel to achieve real speedup due to the caching issues. Just wondering if there exists any specific modification w.r.t the real implementation of the proposed method.

(2) It is possible to have a full theortical time and memory complexity that takes the k also into account? Given that from the language modeling experiments, large bandwidth may affect a lot in the final performance. (20 v.s. 256 context length might not be eligible to be considered as "<<")

(3) More experimental details:
for Figure 5, is there missing one line about high rank with banded attention?

for the language modeling experiments, could the authors provide the sliding window that is used in the main text as well (256 in the appendix)? Moreover, is it able to have the results where the computation of the FMMformer and the transformer have the same compute while the FMMformer could have a longer context window, wondering if that is the more real application we want from the efficient transformer design?

**Main Review:**

The paper is well written and easy to follow.

The proposed method is well-motivated from the physical computation of n-body problems, which seems to be a good extension to the linear kernel attention. It is also developed with the solid support of both theoretical and experimental results for the approximated softmax with certain error bounds. Experiments with multiple random seeds might be preferable to show the significance of the proposed methods. The attached code makes the paper easier for reproduction.

**Time Spent Reviewing:**

16

---

> ### Author Response · Authors · 2021-08-10
> **Rebuttal to the Review of Reviewer zdGe**
>
> Thank you for your valuable review. We address your concerns below.
>
> **Q1. Implementation details: given that local window approach like Big-Bird or Longformer generally needs to have a customized kernel to achieve real speedup due to the caching issues. Just wondering if there exists any specific modification w.r.t the real implementation of the proposed method.**
>
> Answer: We employ the code from https://github.com/idiap/fast-transformers to implement the linear attention. We mimic the code from https://github.com/openai/sparse_attention to implement the local sparse attention with banded matrix in Pytorch.
>
> ------------------------------------------------------------------------------------------------------------------------------
> **Q2. It is possible to have a full theoretical time and memory complexity that takes the k also into account? Given that from the language modeling experiments, large bandwidth may affect a lot in the final performance. (20 v.s. 256 context length might not be able to be considered as "<<")**
>
> Answer: Yes, we will add the full theoretical time and memory complexity in the revision. In particular, for FMMformers with rank $r$ and use a banded matrix with bandwidth $k$, we have:
>
> time complexity: $O(rNDD_v+kND_v)$ and memory complexity: $O((k+r)N)$,
>
> where $D$ is the query and key feature dimension, $D_v$ is the value feature dimension, and $N$ is the total sequence length.
>
> As can be seen in Table 2 in the main text, an FMMformer that uses a banded attention matrix of bandwidth 5 and 1 kernel for far-field (low-rank) attention already improves significantly over the linear transformer. Using more kernels or a better kernel like the fast weight kernel in Table 3 in the Appendix will further improve the performance of the FMMformer.
>
> ------------------------------------------------------------------------------------------------------------------------------
> **Q3. More experimental details: for Figure 5, is there missing one line about high rank with banded attention?**
>
> Answer: We aim to show that the far-field attention can be better modeled as the rank increases; see the discussion in the paragraph titled “Enhancing far-field attention with multi-kernels” in section 4.1. We can add another panel to show the performance of high rank (i.e. rank > 3) with banded attention in our revision.
>
> ------------------------------------------------------------------------------------------------------------------------------
> **Q4. For the language modeling experiments, could the authors provide the sliding window that is used in the main text as well (256 in the appendix)? Moreover, is it able to have the results where the computation of the FMMformer and the transformer have the same compute while the FMMformer could have a longer context window, wondering if that is the more real application we want from the efficient transformer design?**
>
> Answer: The sliding window for the language modeling experiments in the main text also has a length of 256. Yes, making the computation of the FMMformer and the transformer have the same compute while the FMMformer could have a longer context window is a great suggestion to show the advantage of FMMformers. We will include these experiments in our revision.
>
> We appreciate your further feedback.

---

### Official Review · Reviewer_MHtR · 2021-07-15

**Rating:** 5
**Confidence:** 5

**Summary:**

The work proposes FMMformer, which borrows ideas from fast multipole method (FMM) to approximate the quadratic Softmax attention with a linear complexity. Specifically, the approximation consists of a near-field Softmax term (i.e. local attention) and a far-field multi-kernel linear attention. Some theoretical analysis is included to support the idea. The proposed model is evaluated on one synthetic copy task, long range arena and wikitext-103 language modeling. On the three tasks, the proposed model has a better quality compared to linear attention, though the gap between FMMformer and the standard transformer is still significant.

**Limitations And Societal Impact:**

Overall speaking, there won't be any additional negative social impact from the proposed model. In terms of limitation, it would be helpful if authors can discuss that the empirical studies are performed in relatively small scales and in text domain mainly.

**Main Review:**

The key idea of the proposed model is intuitive and well motivated. The theoretical analysis is easy to follow. The practical instantiation of the proposed model is also neat.

However, the empirical study of the work is relatively weak.
- First of all, in terms of baseline, Sparse Transformer (i.e. local + sparse low rank) should be included given its closeness to the proposed model. In addition, the work only considers local attention baselines of bandwidth of at most 20, which could be too limited and won't be able to utilize the full parallelization of GPUs anyway. As the bandwidth increases, local attention could be a strong baseline.
- Secondly, for the 3 tasks in consideration, synthetic copy is overly simplified and long-range arena has relatively large variance and noise. As for wikitext-103, the reported performances of the baselines & proposed model are generally far from the SOTA, weakening the overall credibility of the evaluation.

**Time Spent Reviewing:**

2

---

> ### Author Response · Authors · 2021-08-10
> **Rebuttal to the Review of Reviewer MHtR**
>
> Thank you for your valuable review. We address your comments below.
>
> **Q1. First of all, in terms of baseline, Sparse Transformer (i.e. local + sparse low rank) should be included given its closeness to the proposed model. In addition, the work only considers local attention baselines of bandwidth of at most 20, which could be too limited and won't be able to utilize the full parallelization of GPUs anyway. As the bandwidth increases, local attention could be a strong baseline.**
>
> Answer: The transformer with banded attention only is a special sparse transformer, which has been compared in our paper. We will compare different sparse patterns in Child et al.’s Sparse Transformer in our revised paper. Based on our testing, FMMformer outperforms different sparse transformers on our three tested tasks by a remarkable margin.
>
> We can increase the bandwidth, and based on our test, as the bandwidth increases, the performance of FMMformer becomes better before saturating. Also, using a very large bandwidth in FMMformer can outperform the standard transformer.
>
> We agree with the reviewer that a powerful GPU can allow local attention of larger bandwidth to be computed in parallel. This optimal bandwidth is then device and task-dependent. However, when the bandwidth is large, the quadratic computational and memory complexities then become the bottleneck. Our paper tries to show that by combining local (near-field) attention and non-local (far-field) low-rank linear attention, we can reduce the bandwidth of local attention significantly to avoid this quadratic complexity issue. In training FMMformer, we train both the banded and low-rank parts concurrently.
>
> ------------------------------------------------------------------------------------------------------------------------------
> **Q2. Secondly, for the 3 tasks in consideration, synthetic copy is overly simplified and long-range arena has relatively large variance and noise. As for wikitext-103, the reported performances of the baselines & proposed model are generally far from the SOTA, weakening the overall credibility of the evaluation.**
>
> Answer: We respectfully disagree with this point.
>
> The synthetic copy task is used to show the potential of integration of near-field and far-field attentions and illustrate each part's benefits. The long-range arena benchmark is proposed and has been widely used to validate the performance of (efficient) transformers on tasks that require long-range dependency, see e.g. [1, 2, 3, 4] below.
>
> For wikitext-103, due to the shortage of GPUs in our group, we only compare with the baseline softmax, linear, and fast weight transformers. The improvement of using FMMformer over the baseline is significant; please refer to Table 2 in the paper and Table 3 in the appendix. Note that when using 2 fast weight kernels and Band$_{20}$ provides almost comparable test PPL as the standard transformer (34.71 vs. 34.29), here, FMMformer has linear complexity while the standard transformer has quadratic complexity.
>
> Moreover, our FMMformer can be integrated with better baseline far-field attention models, which can be improved on the fly as the development of better low-rank attention models.
>
> 1. Xiong, Yunyang, Zhanpeng Zeng, Rudrasis Chakraborty, Mingxing Tan, Glenn Fung, Yin Li, and Vikas Singh. "Nystr\" omformer: A Nystr\" om-Based Algorithm for Approximating Self-Attention." AAAI. 2021.
> 2. Peng, Hao, Nikolaos Pappas, Dani Yogatama, Roy Schwartz, Noah A. Smith, and Lingpeng Kong. "Random feature attention." ICLR. 2021.
> 3. Lee-Thorp, James, Joshua Ainslie, Ilya Eckstein, and Santiago Ontanon. "FNet: Mixing Tokens with Fourier Transforms." arXiv preprint arXiv:2105.03824 (2021).
> 4. Yi Tay et al. “Long Range Arena: A Benchmark for Efficient Transformers.” ICLR. 2021.
>
> We appreciate your further feedback.

---

### Official Review · Reviewer_pp7v · 2021-07-16

**Rating:** 4
**Confidence:** 4

**Summary:**

This paper proposed FMMformers, and propose to combine banded attention with low-rank attention in a single model. They conduct experiments on wikitext, copy and long range arena.

**Main Review:**

The combination of different types of attention is not new. Models like BigBird, ETC, long short transformers, luna etc. already proposed this idea. The authors learn to blend the attention types together by a linear combination. I don't think this paper is terribly interesting given this naive composition.

This paper combines banded attention with kernel (low-rank) attention by combining the attention weights before multiplying by V. I think we can think of it as linear/performer transformers with a sprinkle of local attention.  I really think the FMM (fast multipole) angle is **not** main text worthy as it can be just distracting and does not add much value to the understanding of the method.

I think the authors can be upfront that this method is simply add(low rank attention, local_attention). The story does not seem to contribute anything to the understanding of the method and is making things "fancy" unnecessarily. Please consider a more down to earth method if we hope that this method to be more widely adopted. I would appreciate a model name and title that better explains what the method is doing (local+kernel attention) seems to be more appropriate.

Some concerns/weaknesses:
1. The choices of sub-modules "near field" and "far-field" are quite arbitrary.
2. Is the banded matrix just local attention? it seems so. if so, please call it a local attention so it is just less confusing or at least tell us why it is different.
3. Not all models are compared in the long range arena even though the results are already in the original paper.
4. is language modeling the right choice of experiments for long range transformers?






**Time Spent Reviewing:**

0.5

---

> ### Author Response · Authors · 2021-08-10
> **Response to Your Terribly Unethical and Wrong Review.**
>
> Thank you for spending 0.5 hour reviewing our paper, which might be the reason that you misunderstood our paper and made wrong comments.
>
> Before addressing your wrong comments, we regret to point out that your review is wrong and want to stress that your review is terribly unethical, which makes us fear you are not qualified for being a NeurIPS reviewer or perhaps even doing any peer review. In particular, you falsify that our work is not new by comparing two nonexisting papers by the NeurIPS deadline. Note that NeurIPS paper submission deadline was  **May 28th, 2021**. However, the Long-Short Transformer paper (https://arxiv.org/abs/2107.02192) was first released on arXiv on **Jul 5th, 2021 and updated on July 27th**. Also, the Luna paper (https://arxiv.org/pdf/2106.01540.pdf) was first released on **June 3rd, 2021**. Let alone if our work is related to the above papers or not which will be addressed in our response to your review, what you did is highly unethical. **We are considering reporting your unethical behavior to the program chairs.** We do not know who you are, but your falsification can even damage the reputation of the authors of the above two papers.
>
> Now we turn to respond to your wrong comments.
>
> **1. You falsify that our work is not new by saying that models like BigBird, ETC, long short transformers, luna etc. already proposed the idea of combining different types of attention.**
>
> Answer: We disagree with your wrong and unethical comment.
> **Long-Short transformers and Luna papers did not exist by the NeurIPS submission deadline.** Falsifying our work is not new by comparison to those work is unethical. We can also say Long-Short transformers and Luna is not new based on your mentality. **We can’t help questioning the purpose of the reviewer when using these two works to downgrade the novelty of our paper.**
>
> Below we will explain the novelty of our method and compare it to the works that the reviewer mentioned.
>
> BigBird combines only sparse attentions with different sparse structures. The ETC divides the input into the global input and long input and uses local sparsity to reduce the quadratic scaling of the attention mechanism. Long-short transformers combine long-range and short-range attention, but use dynamic projection to capture long-range attention. Luna approximates softmax attention with two nested linear attention functions. In contrast to those transformers, our FMMformers use banded attention matrix to capture local/near-field attention and linear attention to capture far-field attention.
>
> -------------------------------------------------------------------------------------------------------------------------------
> **2. You commented that “This paper combines banded attention with kernel (low-rank) attention by combining the attention weights before multiplying by V. I think we can think of it as linear/performer transformers with a sprinkle of local attention. I really think the FMM (fast multipole) angle is {\bf not} main text worthy as it can be just distracting and does not add much value to the understanding of the method. I think the authors can be upfront that this method is simply add(low rank attention, local_attention). The story does not seem to contribute anything to the understanding of the method and is making things "fancy" unnecessarily. Please consider a more down to earth method if we hope that this method to be more widely adopted. I would appreciate a model name and title that better explains what the method is doing (local+kernel attention) seems to be more appropriate.”**
>
> Answer: We disagree with your comment. This comment shows you do not understand our work, which we can understand since you only spent 0.5 hour for the review. You were trying to trivialize our work due to your misunderstanding of our work.
>
> We first numerically showed that the self-attention matrix has the low-rank structure; in particular, when excluding a banded matrix from the self-attention matrix (see Section 2.2). The low rank structure of the self-attention has also been noticed in the Linformer paper (https://arxiv.org/pdf/2006.04768.pdf). The low rank structure of the self-attention matrix after excluding a banded matrix exactly falls into the algebraic viewpoint of the fast multipole method, which also provides the theoretical error bound of using low rank combined with sparse matrices to approximate dense matrix that has low rank structure after excluding a banded matrix (see Section 2.1). The algebraic viewpoint of FMM serves as the theoretical motivation for our FMMformer.
>
> We believe FMMformer is a reasonable name for our model since the key idea of FMM is using sparse and low rank models for describing near-field and far-field interaction, and our proposed efficient transformer is motivated by FMM.
>
> -------------------------------------------------------------------------------------------------------------------------------
> **3. You commented that “The choices of sub-modules "near field" and "far-field" are quite arbitrary.”**
>
> Answer: They are not arbitrary. In FMMformer, near-field attention has to be a banded matrix; far-field attention has to be a low rank matrix. The choice of bandwidth and low rank matrix has some flexibility, which should be considered as an advantage of FMMformer.
>
> How to design optimal near-field and far-field attention is our future work.
>
> -------------------------------------------------------------------------------------------------------------------------------
> **4. Is the banded matrix just local attention? it seems so. if so, please call it a local attention so it is just less confusing or at least tell us why it is different.**
>
> Answer: We are happy to also call it local attention; while we think near-field attention is a reasonable name considering FMMformer is motivated by FMM.
>
> -------------------------------------------------------------------------------------------------------------------------------
> **5. Not all models are compared in the long range arena even though the results are already in the original paper.**
>
> Answer: We report the baseline results from [1]. Results for reformer, linformer, performer, and Nystromformer are also reported in [4]. All of them are worse than our FMMformer’s results.
>
> 1. Xiong, Yunyang, Zhanpeng Zeng, Rudrasis Chakraborty, Mingxing Tan, Glenn Fung, Yin Li, and Vikas Singh. "Nystromformer: A Nystrom-Based Algorithm for Approximating Self-Attention." AAAI. 2021.
>
> -------------------------------------------------------------------------------------------------------------------------------
> **6. Is language modeling the right choice of experiments for long range transformers?**
>
> Answer: Yes, it is, with a proper dataset. The language modeling experiments are done on WikiText-103 which has sufficiently long contextual dependencies with the contextual text blocks of about 3600 words. Also, this dataset has been used widely, e.g. "Schlag et al. Linear Transformers Are Secretly Fast Weight Programmers, ICML 2021" to study the effectiveness of efficient transformers.
>
> We appreciate your further feedback and are willing to address your further reasonable comments.

---

### Official Review · Reviewer_bVJZ · 2021-07-19

**Rating:** 4
**Confidence:** 4

**Summary:**

A fast and memory-efficient transformer architecture based on a sparse attention mechanism inspired by the fast multipole method (FFM). It decomposes attention matrices into the sum of a banded matrix (capturing near-range interactions) and a low rank matrix (capturing long-range interactions). This low rank matrix is itself expressed as a sum of rank 1 matrices, each characterized by a given feature map. Experiments on the LRA benchmark and WikiText-103 show promising results in terms of test accuracy/perplexity Vs training/inference speedups.

**Limitations And Societal Impact:**

Limitations are discussed, although they are a bit scattered throughout -- would be helpful to synthesize them in conclusion.

**Main Review:**

**Overall appreciation (originality, quality, clarity, and significance)**
- Originality: Adapts the intuition from the fast multipole method to sparsify attention maps, and leverage several tricks (e.g., kernel trick for linear transformers from ​​Katharopoulos et al.) to obtain a simple and intuitive formulation.
- Quality: Method is well motivated and there are a few promising experimental results. However the experimental design has significant gaps (see suggestions below).
- Clarity: Easy to read overall. There are some subsections that felt unnecessary / not concisely written (see below).
- Significance: Although difficult to draw conclusions given the shortcomings of current experiments, the idea appears to have potential. The duality of local attention vs long-range attention controlled by two separate components is compelling. Additional guidance/analyses on relative performance tradeoffs between widening the banded matrix Vs adding more feature maps in low rank matrix, as well as on the way to choose these feature maps would be very valuable.

**Clarifying questions**
- Fig 4 & 5: Are these showing cross entropy loss on the training set or validation set? If on the training set then results are not surprising (you are training networks with strictly more parameters). If on validation set, then the language (line 201) is a bit misleading

**Suggestions**
- Section 2.1 - This subsection could be made more concise: several definitions and propositions are introduced but subsequently not used anywhere (could be moved to appendix)
- Section 3 - Guidance on how to select / craft these feature maps would strengthen your paper
- Fig 4 & 5 - It appears you have not yet converged on the longer sequences experiment (length=512) which makes your conclusion on line 202 not clear. Additionally, I would keep consistent the y-axis scales between the three columns: as such they are a bit hard to compare across.
- Section 4 - This section is lacking a speed vs acc/perplexity analysis (in particular on LRA). Currently, section 4.1 has . Would be good to have in table format for speedups to appreciate actual gains. FIg.
- Section 4 - It would be very helpful to perform a thorough ablation analysis that shows relative trade-offs (on accuracy/perplexity, speed, memory) from increasing the banded matrix bandwidth Vs increasing the number of feature maps considered for the low rank matrix.
- Section 4 - There are many sparse attention transformer architectures out there (e.g., Performer, Linformer). It is important to compare FMMformer to these other methods (besides linear transformers) to understand relative performance trade-offs
Section 4 - Would recommend keeping things more consistent in terms of choice of parameters across experiments (number of kernels and bandwidth in particular)

**Minor points**
- Line 72/73: “and has been regarded as one of the top 10 algorithms in scientific computing in the 20th century” → unnecessary, would suggest to drop this sentence
- Line 86-87: “For the illustration purpose” → “For illustrative purposes”
- Line 94: Define C (e.g., positive constant)
- Line 153: ‘using the fact that LV= …” → would drop -- you have the same identity just above
- Line 159: Would indicate what these assumptions are here
- Line 174/175: “It is easy to check that [...] linearly independent for almost all x” → Add proof in appendix
- Line 220: “capturing long-term dependency’ → “long-range dependencies”


**Time Spent Reviewing:**

7

---

> ### Author Response · Authors · 2021-08-10
> **Rebuttal to the Review of Reviewer bVJZ**
>
> Thank you for your valuable review. We address your concerns point-by-point below.
>
> **Q1. Fig 4 & 5: Are these showing cross entropy loss on the training set or validation set? If on the training set then results are not surprising (you are training networks with strictly more parameters). If on validation set, then the language (line 201) is a bit misleading.**
>
> Answer:
> Fig. 4 & 5 show the cross-entropy loss on the training set, which we follow the same experimental design as in “Katharopoulos et al., Transformers are RNNs: Fast Autoregressive Transformers with Linear Attention, ICML 2020”.
>
> In Fig. 4, our FMMformers with different ranks only have one additional parameter at each layer for blending near-field and far-field attention. The model we use has four attention layers in total, so our FMMformers have only 4 additional parameters compared to the baseline linear transformer. Furthermore, more parameters do not guarantee faster convergence, which may lead to small training loss eventually. A model with more parameters might converge slower initially but then achieve a better training loss. Fig. 4 & 5 show that all FMMformers converge faster than the baseline linear transformer throughout the training. Again, here we followed exactly the same experimental design and want to convey the same message as in “Katharopoulos et al., Transformers are RNNs: Fast Autoregressive Transformers with Linear Attention, ICML 2020”, but for different efficient transformers.
>
> ------------------------------------------------------------------------------------------------------------------------------
> **Q2. Section 2.1 - This subsection could be made more concise: several definitions and propositions are introduced but subsequently not used anywhere (could be moved to appendix).**
>
> Answer:
> Section 2.1 briefly discusses the algebraic viewpoint of the fast multipole method, which serves as the theoretical motivation of our FMMformer. In particular, Section 2.1 explains that the combination of sparse and low-rank attention is a good approximation to the full softmax attention, provided the softmax-attention is low rank after excluding a banded matrix which is numerically backed up in Section 2.2 (also observed in the Linformer paper [1], https://arxiv.org/pdf/2006.04768.pdf).
>
> We will try to compress this section to make it more concise in our revision.
>
> [1] Wang, Sinong, Belinda Z. Li, Madian Khabsa, Han Fang, and Hao Ma. "Linformer: Self-attention with linear complexity." arXiv preprint arXiv:2006.04768 (2020).
>
> ------------------------------------------------------------------------------------------------------------------------------
> **Q3. Section 3 - Guidance on how to select / craft these feature maps would strengthen your paper.**
>
> Answer:
> Thanks for your suggestion. Intuitively, the feature maps should be chosen to be linearly independent to each other to make the low-rank attention more expressive. Based on our testing, if two feature maps differ more, the improvement becomes more significant, inspiring us to consider orthogonal feature maps in future work. This design is nontrivial since there is also a stability issue; in our experiments, we noticed that the training may fail if the feature maps are not well selected. We will emphasize this in our revision.
>
> ------------------------------------------------------------------------------------------------------------------------------
> **Q4. Fig 4 & 5 - It appears you have not yet converged on the longer sequences experiment (length=512) which makes your conclusion on line 202 not clear. Additionally, I would keep consistent the y-axis scales between the three columns: as such they are a bit hard to compare across.**
>
> Answer: In Fig 4 & 5, we ran the same number of iterations as that did in the Linear transformer paper (https://linear-transformers.com/), which is a benchmark of our work. We have rerun experiments for sequence length 512 in Fig. 4 & 5 with 20K gradient steps. After convergence, we still see the benefits of adding near-field attention over the baseline linear (far-field) attention as claimed in line 202. We have unified the y-axis scales in our revision as you suggested.
>
> ------------------------------------------------------------------------------------------------------------------------------
> **Q5. Section 4 - This section is lacking a speed vs acc/perplexity analysis (in particular on LRA). Currently, section 4.1 has. Would be good to have in table format for speedups to appreciate actual gains.**
>
> Answer: Following the reviewer’s suggestion, we have done a speed vs. acc/perplexity analysis on the LRA benchmark. On average, we observed that FMMformers with 1 and 2 kernels achieve similar speed as the linear and sparse transformers with bandwidth 5 (Band$_{5}$). We summarize our analysis in Table 1 below.
>
> Table 1: Speed at train time (second/sample)
>
> |        | Listops     | Text     | Retrieval     | Image     | Pathfinder     | Avg
> | :------------- | :----------: | -----------: | ------: | -----------: | -----------: | -----------: |
> | Softmax | 0.0069   | 0.0022    | 0.0417    | 0.0009    | 0.0018    | 0.0107
> | Linear   | 0.0014 | 0.0009 | 0.0123 | 0.0003 | 0.0005 | 0.0031
> | Band$_{5}$   | 0.0012 | 0.0008 | 0.0120 | 0.0002 | 0.0003 | 0.0029
> | FMMformer (1-kernel + Band$_{5}$)   | 0.0015 | 0.0009 | 0.0123 | 0.0003 | 0.0006 | 0.0031
> | FMMformer (2-kernel + Band$_{5}$)   | 0.0015 | 0.0010 | 0.0123 | 0.0004 | 0.0006 | 0.0032 |
>
> ------------------------------------------------------------------------------------------------------------------------------
> **Q6. Section 4 - It would be very helpful to perform a thorough ablation analysis that shows relative trade-offs (on accuracy/perplexity, speed, memory) from increasing the banded matrix bandwidth Vs increasing the number of feature maps considered for the low-rank matrix.**
>
> Answer: We will add this ablation in our revised paper. Also, Figures 4, 5, and 6 in our paper show this ablation analysis for the copy task.
>
> ------------------------------------------------------------------------------------------------------------------------------
> **Q7. Section 4 - There are many sparse attention transformer architectures out there (e.g., Performer, Linformer). It is important to compare FMMformer to these other methods (besides linear transformers) to understand relative performance trade-offs Section 4 - Would recommend keeping things more consistent in terms of choice of parameters across experiments (number of kernels and bandwidth in particular).**
>
> Answer:
> In Table 3 in the Appendix, we compare the FMMformer with fastweight transformers on the language modeling task, which has been shown to have better performance than both performers and linformers [1]. Also in [1], the authors show that the baseline linear transformer we used in the main text has better performance than the performer across tasks. In addition, on the LRA benchmark, our reported average accuracies for FMMformers in Table 1 in the main text (59.51% for 1 kernel and 60.74% for 2 kernels) are better than the accuracy of performer and linformer reported in [2], which are 53.63% and 55.59%, respectively.
>
> [1] Schlag, Imanol, Kazuki Irie, and Jürgen Schmidhuber. "Linear Transformers are secretly fast weight programmers." In International Conference on Machine Learning, pp. 9355-9366. PMLR, 2021.
>
> [2] Xiong, Yunyang, Zhanpeng Zeng, Rudrasis Chakraborty, Mingxing Tan, Glenn Fung, Yin Li, and Vikas Singh. "Nystr\" omformer: A Nystr\" om-Based Algorithm for Approximating Self-Attention." AAAI. 2021.
>
> ------------------------------------------------------------------------------------------------------------------------------
> **Q8.  1)Line 72/73: “and has been regarded as one of the top 10 algorithms in scientific computing in the 20th century” → unnecessary, would suggest to drop this sentence. 2) Line 86-87: “For the illustration purpose” → “For illustrative purposes”. 3) Line 94: Define C (e.g., positive constant). 4) Line 153: ‘using the fact that LV= …” → would drop -- you have the same identity just above. 5) Line 159: Would indicate what these assumptions are here. 6) Line 174/175: “It is easy to check that [...] linearly independent for almost all x” → Add proof in appendix. 8) Line 220: “capturing long-term dependency’ → “long-range dependencies”.**
>
> Answer: Thank you so much for pointing these out. We will address them in our revision.
>
>
> ------------------------------------------------------------------------------------------------------------------------------
>
> We appreciate your further feedback.

---

> > ### Comment · Reviewer_bVJZ · 2021-08-30
> > **Thank you for the response**
> >
> > Dear authors,
> >
> > Thank you very much for the detailed response and additional analysis.
> >
> > Q1 & Q4 -- I understand the analysis in Katharopoulos et al. to be reported on the validation set (based on the trajectory of the softmax curve on Fig2). Do let me know if you think this is incorrect. Either way, I do believe the most sensible thing to do here is to report the cross entropy on the validation set.
> >
> > Q5 -- Thank you for the new analysis. Minor point: I would recommend using a different unit or showing at least 2 significant digits everywhere
> >
> > Q7 -- Performer and Linformer were given as two examples -- but there are several other relevant baselines, as referenced in your related work section. The main point was that just comparing the performance to linear transformer is not sufficient, since it is a particular base of your method with rank 1 kernel and no local attention. I would highly encourage to add several of these baselines in your own tables, detailing the relative trade-offs between these methods in terms of speedups/FLOPS, memory gains, convergence speed, test accuracy/perplexity. It’s not fair to the reader to have to dig these numbers across papers otherwise.

---

> > > ### Author Response · Authors · 2021-08-30
> > > **Thank you!**
> > >
> > > Dear Reviewer,
> > >
> > > Thank you for your further feedback and valuable suggestions.
> > >
> > > ------
> > >
> > > **Q1. I do believe the most sensible thing to do here is to report the cross-entropy on the validation set.**
> > >
> > > Answer: Thank you for your suggestion. We will report the cross-entropy on the validation set in our revised paper. We do have these results.
> > >
> > > -----
> > >
> > > **Q2. I would recommend using a different unit or showing at least 2 significant digits everywhere.**
> > >
> > > Answer: We will report at least 2 significant digits in our revised paper.
> > >
> > > -----
> > >
> > > **Q3. I would highly encourage to add several of these baselines in your own tables, detailing the relative trade-offs between these methods in terms of speedups/FLOPS, memory gains, convergence speed, test accuracy/perplexity. It’s not fair to the reader to have to dig these numbers across papers otherwise.**
> > >
> > > Answer: For the ease of readers, we will add these comparisons in our revised paper as you suggested.
> > >
> > > -----
> > >
> > > Based on your further feedback, we believe we have addressed your concerns about our work. We appreciate your further feedback before the end of the discussion period. Thanks, again.

---

> > > > ### Comment · Reviewer_bVJZ · 2021-08-31
> > > > **Final comments**
> > > >
> > > > Dear authors,
> > > >
> > > > Thank you for the additional responses. To clarify where I stand after the rebuttal: while you provided a response to the different points that were raised and alleviated some of the concerns on the experiment design, my overall appreciation of the paper remains unchanged at this point (reject) given the significant rework needed based on our discussion. To recap the most important points that need to be changed:
> > > > 1. The experimental section is lacking thorough comparisons with baselines, on a broader set of metrics (as per my prior message + Q7)
> > > > 2. Since your method sparsifies attention maps in order to yield speedups & memory gains, it is critical to illustrate the relationship between these gains and test accuracy/perplexity across all experiment settings. The table you provided in (Q5) is a good step in that direction. The same needs to be done for the language modeling exp. You need to convince the reader that your method does help achieve a better speed Vs test perplexity tradeoff than baselines
> > > > 3. Building on point #2, you should show how different values of the main hyperparameters for your method impact that trade-off (Q6). This is critical to understand from a practical standpoint, and more convincing if done on the LRA and language modeling experiments (than on the copy task experiment which is a toy setting)
> > > > 4. The background section (2.1 in particular) needs to be significantly streamlined (Q2)
> > > >
> > > > This represents a major overhaul of the current version of the paper, hence why my overall appreciation of the paper remains unchanged.

---

> > > > > ### Author Response · Authors · 2021-09-01
> > > > > **Thanks**
> > > > >
> > > > > Thanks for your feedback.

---

### Decision · Program_Chairs · 2021-09-27

**Decision:**

Accept (Poster)

**Comment:**

This paper proposed a fast and memory-efficient transformer approximation, inspired by the fast multiple method (FFM), which combined the sparse and low-rank approximation. The method is well-motivated and easy-to-follow. The authors demonstrated the benefits of the proposed neural architecture empirically on synthetic data, LRA benchmark, and language modeling on WikiText-103.

There are several existing work on combining sparse transformer with low-rank transformer,  as pointed by the reviewers (pp7v and MHtR), however, since these are the co-current work, the novelty of the submission should not be discounted. Actually, as opposite. it actually strengthens the significancy of the method. I personally like Sec 2.1 to motivate the FMM.

The major criticism from most of the reviewers actually lies in the empirical experiment part, which makes the claim of the paper relatively weak. In my opinion, these suggestions are reasonable. For example, the comprehensive comparison with more baselines, detailed ablation study on the trade-off between accuracy, speed, memory vs. architecture parameters, and more practical tasks on real-world benchmarks should be included. The authors rebuttal addressed some of them, but some of them are still missing.

In sum, the idea and new architecture proposed by this paper is novel and well-motivated, however, the benefits of this architecture is not well empirically justified. I hope in the final version, the authors can take the reviewers' comments into account to improve the weakness of the paper.